# Development of an Electrochemical Sensor Based on Nanocomposite of Fe₃O₄@SiO₂ and Multiwalled Carbon Nanotubes for Determination of Tetracycline in Real Samples

**Edna Ferreira Amaral, Daniela Nunes da Silva, Maria Cristina Silva and Arnaldo César Pereira \***

Departamento de Ciências Naturais, Campus Dom Bosco, Universidade Federal de São João del Rei (UFSJ), Praça Dom Helvécio 74, Fábricas, São João del Rei, MG 36301-160, Brazil; faedna17@gmail.com (E.F.A.); daniela.silva.quimica@gmail.com (D.N.d.S.); crisiria@ufsj.edu.br (M.C.S.)
**\*** Correspondence: arnaldo@ufsj.edu.br

**Abstract:** In this work, an electrochemical sensor (GCE/MWCNT/Fe₃O₄@SiO₂) based on a composite of multiwalled carbon nanotubes (MWCNT) and an Fe₃O₄@SiO₂ (MMN) nanocomposite on a glassy carbon electrode (GCE) was developed for the detection of tetracycline (TC). The composite formed promoted an increased electrochemical signal and the stability of the sensor, combining its individual characteristics such as high electrical conductivity and large surface area. The composite material was characterized by X-ray diffraction (XRD), Fourier transform infrared spectroscopy (FTIR), Mössbauer spectroscopy, and scanning electron microscope (SEM). The adsorptive stripping differential pulse voltammetry (AdSDPV) promoted better performance for the electrochemical sensor and greater sensitivity for TC detection. Under optimized conditions, the currents increased linearly with TC concentrations from 4.0 to 36 $\mu$mo L$^{-1}$ (0.997) and from 40 to 64 $\mu$mo L$^{-1}$ (0.994) with detection and quantification limits of 1.67 $\mu$mo L$^{-1}$ and 4.0 $\mu$mo L$^{-1}$, respectively. The sensor was applied in the analysis of milk and river water samples, obtaining recovery values ranging from 91–117%.

**Keywords:** electrochemical sensor; multiwalled carbon nanotubes; nanocomposite of Fe₃O₄@SiO₂; tetracycline; water samples; milk samples

## 1. Introduction

Tetracyclines (TCs) are a class of broad-spectrum antibiotics with high activity against various infections caused by Gram-positive and Gram-negative bacteria as well as a variety of organisms including mycoplasma and chlamydia [1]. Currently, there are more than twenty TC derivatives available, among which, tetracycline, chlortetracycline, doxycycline, and oxytetracycline are the most widely used [2] in the treatment of diseases in humans and veterinary medicine as a food additive to treat and prevent bacterial infections and as growth promoters [3]. Given the high consumption of these antibiotics, TC residues have been detected in aquatic environments [4] and in foods of animal origin such as milk [5] and honey [6], which represents a serious threat to human health, since it contributes to the development of antimicrobial resistance and the appearance of allergic reactions [7].

Thus, it is of great importance to developing a sensitive, simple, and low-cost methodology to detect TC residues in food, biological, and environmental samples. In recent years, several methodologies have been developed for the determination of TC including chromatographic methods [8,9], capillary electrophoresis [10,11], fluorescence [12], and electrochemical detection. Electrochemical sensors have several advantages such as low cost, simplicity, high sensitivity, high selectivity, and the possibility of on-site analysis, among others [13].

Wong et al. developed an electrochemical sensor modified with multiwalled carbon nanotubes and graphene oxide for tetracycline detection. This sensor showed a linear response for tetracycline from $2.0 \times 10^{-5}$ to $3.1 \times 10^{-4}$ mol L$^{-1}$ and a detection limit of

$3.6 \times 10^{-7}$ mol L$^{-1}$ by using adsorptive stripping differential pulse voltammetry (AdS-DPV) [14].

The possibility of developing large-scale methodologies with low-cost materials such as carbon-based materials has further boosted research for the development of electrochemical sensors and biosensors. Several studies in the literature have shown that sensors modified with multiwalled carbon nanotubes (MWCNT) and a nanocomposite of Fe$_3$O$_4$@SiO$_2$ [15,16] promote increased selectivity, stability, and sensitivity.

The carbon nanotubes have properties such as high surface area, excellent electrical conductivity, mechanical strength, and high chemical stability [17,18]; in sensors, it can be used to promote the transfer of electrons between electroactive species and the electrode surface [19,20].

Magnetic iron-oxide-based nanoparticles, known for their applications in magnetic separation in the sensory field, have been used due to their catalytic properties [21–23].

Fe$_3$O$_4$@SiO$_2$ magnetic nanoparticles with high surface area can be integrated into sensors in a variety of ways, for example, through contact between the metallic magnetic nanoparticle and electrode surface, or through the transport of a redox-active species to the electrode surface, or by a thin film on the electrode surface [24]. Synergistic effects between nanocomposites of Fe$_3$O$_4$@SiO$_2$ and MWCNTs can improve the capacities of the electrochemical sensor [25].

In the present work, we report on a glassy carbon electrode (GCE) modified with multi-walled nanotubes carbon and a nanocomposite of Fe$_3$O$_4$@SiO$_2$ (GCE/MWCNT/Fe$_3$O$_4$@SiO$_2$) for the determination of tetracycline (TC) in raw milk and river water samples.

## 2. Materials and Methods

### 2.1. Reagents and Solutions

All chemical reagents used in this work were of analytical purity. The aqueous solutions were prepared with deionized water by a Millipore Milli-Q system (18.2 MΩ·cm, 25 °C). MWCNTs were acquired from Nanocyl. HCl was obtained from Dinâmica. NaOH, FeSO$_4$·7H$_2$O, KNO$_3$, KOH, and APTES (3-aminopropyltriethoxysilane) were obtained from Sigma Aldrich (St. Louis, MO, USA). KCl, CH$_3$COOH, and NaH$_2$PO$_4$·7H$_2$O were obtained from Synth. Ammonium hydroxide was purchased from Quemis. Tetraethoxysilane (TEOS) was purchased from Merck. The stock solution of tetracycline $1.0 \times 10^{-3}$ mol L$^{-1}$ was prepared in 5.0 mL of the HCl $1.0 \times 10^{-3}$ mol L$^{-1}$.

### 2.2. Synthesis of Nanocomposites of Fe$_3$O$_4$@SiO$_2$

The nanocomposite of Fe$_3$O$_4$@SiO$_2$ (MMN) was synthesized according to the literature [26]. The steps of synthesis are illustrated in Figure 1A,B.

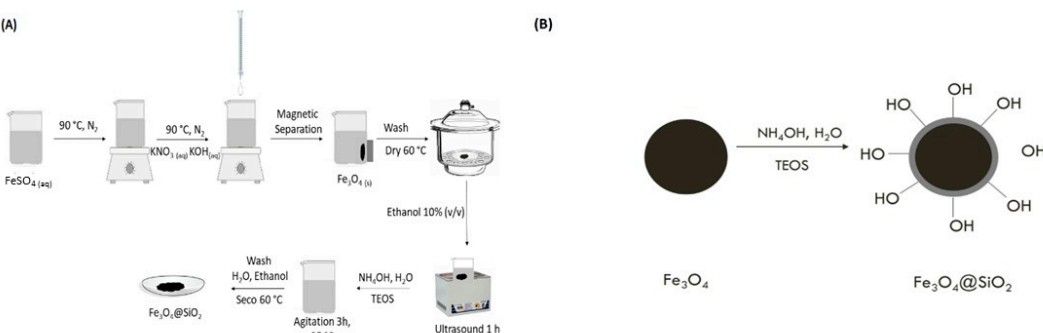

**Figure 1.** (**A**) Co-precipitation synthesis process of silica-coated magnetite. (**B**) Illustrative diagram of the magnetic core coating.

### 2.3. Characterization of the Modifiers

The spectra of materials by FTIR were obtained using a Spectrum GX Perkin Elmer spectrometer with transmission mode ranging from 400 to 4000 cm$^{-1}$ with the accumulation of 32 spectra and resolution of 2 cm$^{-1}$; the KBr (1%) pellet technique was used to characterize the materials. The scanning electron micrographs were obtained by microscopy using JEOL model JSM 300. X-ray diffractograms were obtained by an X'pert Pro X-ray diffractometer, with Cu K$_\alpha$ radiation. The spectra Mössaubauer were obtained at 298 K with a source of 57 Co/Rh in the range of $\pm 10 \times 10^{-3}$ m s$^{-1}$ in 1024 channels in a SEE Co model W302. A $\alpha$-Fe was used as a calibration standard.

### 2.4. Point of Zero Charge of $Fe_3O_4$@$SiO_2$ Nanocomposite

The point of zero charge was determined through an experiment (in duplicate), in which the pH of the sorbent solution was changed by five values (2, 4, 6, 8 and, 10), aiming to find the pH at which the charge in the material surface is equal to zero. A total of 25 mg of the MMNs were dispersed in an aqueous solution of sodium chloride (NaCl) 0.10 mol L$^{-1}$ under five different conditions of initial pH, ranging from 2.0 to 10.0. The adjustment was performed with aqueous hydrochloric acid solutions (0.10 mol L$^{-1}$ and 0.05 mol L$^{-1}$ and NaOH (0.10 mol L$^{-1}$). After 24 h of stirring, the pH value was measured again. The result was interpreted through the initial pH curve versus the final pH. The point of zero charge corresponds to the point where the final pH is equivalent to the initial pH.

### 2.5. Apparatus

The electrochemical measurements were carried out with the use of an Autolab PGSTAT12 potentiostat/galvanostat coupled to a microcomputer containing the NOVA 1.11 and 1.8 software. All measurements were performed in a configuration of three electrodes such as the working electrode of a modified glassy carbon electrode (GCE), Ag/AgCl/KCl 3.0 mol L$^{-1}$ as the reference electrode, and as an auxiliary electrode, a platinum wire. The operational parameters of each technique were as follows: Adsorptive differential pulse voltammetry (AdSDPV)—modulation time range: 1 s; modulation time: 10 s; accumulation potential: −0.020 V; accumulation time: 15 s, and accumulation time range: 1 s. Differential pulse voltammetry (DPV)—amplitude 0.1 V; scan rate 0.030 Vs$^{-1}$. Square wave voltammetry (SWV)—amplitude: 0.030 V; frequency: 25 Hz.

### 2.6. Preparation of GCE/MWCNT/$Fe_3O_4$@$SiO_2$ Modified Electrode

The cleaning of GCE was performed with an alumina suspension, followed by washing with deionized water. After that, the electrode was electrochemically cleaned by twenty consecutive cycles in cyclic voltammetry (CV) in 0.5 mol L$^{-1}$ sulfuric acid, varying the potential from −1.4 to +1.4 V vs. Ag/AgCl at a scan rate of 0.025 V s$^{-1}$.

In the last step, 5 µL of MWCNT (2.50 mg mL$^{-1}$) and 5 µL of MMN (2.50 mg mL$^{-1}$) were dispersed in dimethylformamide (DMF) and deposited on the surface of the GCE. Then, the electrode was left to dry at 60 °C for approximately 10 min for complete evaporation of the solvent.

### 2.7. Evaluation of Experimental Parameters

The amount of MWCNTs was changed in the range of 1.0 to 3.0 mg (dispersed in 1.0 mL of DMF). After this stage, the amount of MWCNT was fixed and the amount of MMN was changed in the range of 0.5 to 2.0 mg (dispersed in 1.0 mL DMF). After the optimization of the working electrode, the influence of pH in the electrolytic medium was studied in intervals from 2.0 to 7.0 using the solution of KCl 0.1 mol L$^{-1}$. After setting the best pH value, the types of support electrolyte (hydrochloric acid, Britton–Robinson, phosphate, and acetate) were evaluated. All studies were performed in the presence of TC 22 µmo L$^{-1}$.

The selectivity of the GCE/MWCNT/Fe$_3$O$_4$@SiO$_2$ sensor was evaluated in the presence of possible interferents such as doxycycline, amoxicillin, and diclofenac. All solutions were prepared at a concentration of $1.0 \times 10^{-3}$ mol L$^{-1}$.

*2.9. Application in Real Samples*

The GCE/MWCNT/Fe$_3$O$_4$@SiO$_2$ sensor was applied in the analysis of river water and raw milk samples. The river water was collected in the city of São João del Rei, in the interior of Minas Gerais State (Brazil). The samples were collected according to the sampling protocol [27].

The raw milk samples were collected in two different locations: São João del Rei and the Elvas District (Minas Gerais, Brazil). To prepare the milk sample, it was necessary to separate some constituents of milk (fat, protein) that could interfere with TC detection. For milk pre-treatment, the proposed methodology in the literature was used [28].

Then, the pre-treated samples were enriched with known concentrations of TC. The recovery rates of the developed method were obtained through the analytical curve.

## 3. Results and Discussion

### 3.1. Characterization of the Modifiers

The SEM images obtained for the modifiers are shown in Figure 2. For the MWCNTs, a tubular structure was observed. In Figure 2B, the Fe$_3$O$_4$@SiO$_2$ has a surface made up of spherical, agglomerated particles due to the magnetic core (Fe$_3$O$_4$). Finally, the composite MWCNT/Fe$_3$O$_4$@SiO$_2$ (Figure 2C) shows that these materials were well distributed.

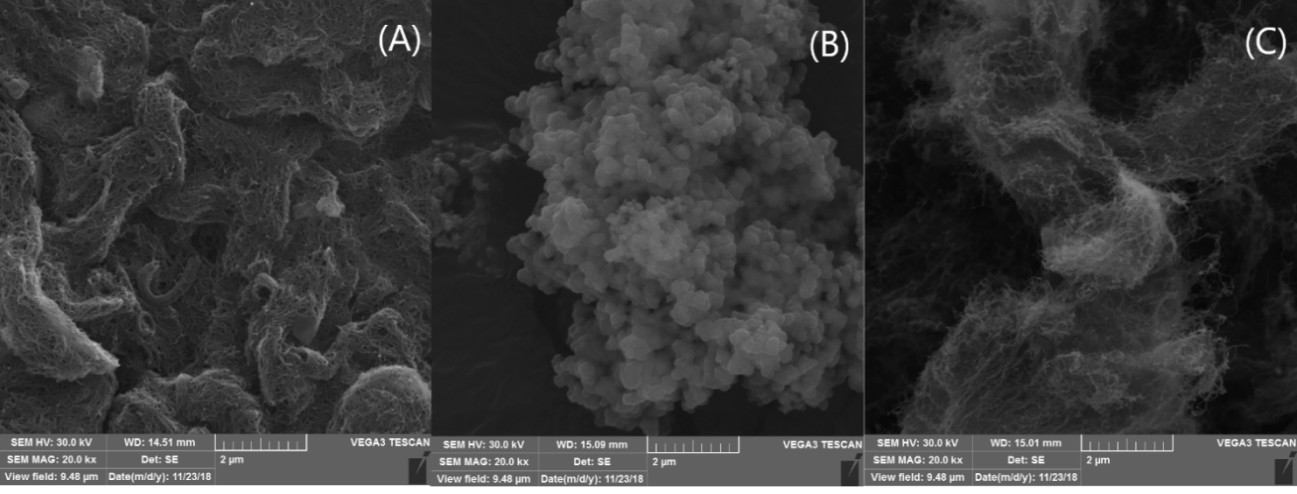

**Figure 2.** SEM. (**A**) MWCNT (**B**) Fe$_3$O$_4$@SiO$_2$, and (**C**) MWCNT/Fe$_3$O$_4$@SiO$_2$.

Another technique used for characterization was FTIR (Figure 3A). The FTIR spectrum of MWCNT showed bands of deformation of the bond C–H$_2$ methylene group around 2926 and 2869 cm$^{-1}$ [29]. A broad band around 3400 cm$^{-1}$ may be associated with the stretching vibration mode of the O–H bond of functional groups that are on the surface of the MWCNTs. Stretch bands of the C=C bonds around 1640 cm$^{-1}$, 1497 cm$^{-1}$, and 1440 cm$^{-1}$, characteristics of aromatic compounds, were also observed [30]. In the Fe$_3$O$_4$@SiO$_2$ spectra, the bands at around 570 cm$^{-1}$ and 449 cm$^{-1}$ can be assigned to the stretching and angular deformation of the Fe–O bond in the magnetic core, respectively.

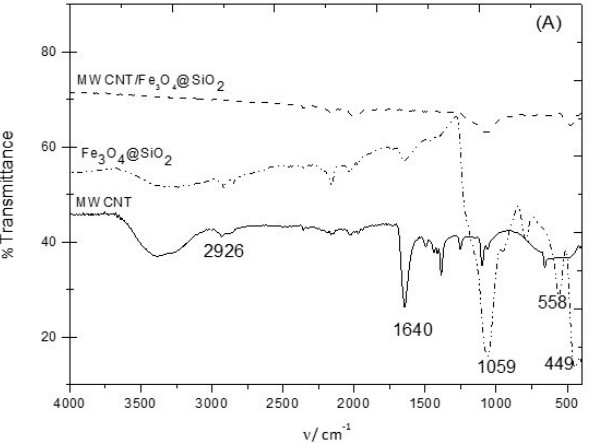 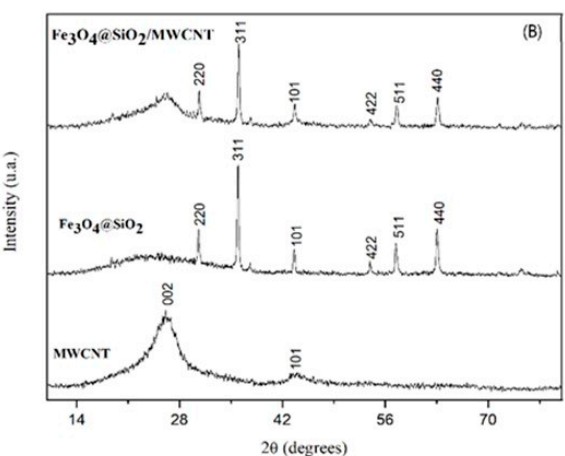

**Figure 3.** (**A**) FTIR spectra of MWCNT; $Fe_3O_4$@$SiO_2$, and MWCNT/$Fe_3O_4$@$SiO_2$. (**B**) X-ray diffractograms of MWCNTs; $Fe_3O_4$@$SiO_2$ and MWCNT/$Fe_3O_4$@$SiO_2$.

The bands at 922 and 862 $cm^{-1}$ are due to the stretching of the bonds (Si–O) [31] and the bands around 1100 to 1000 $cm^{-1}$ are related to the bonds (Si–O–Si) [32]. The difference between the $Fe_3O_4$@$SiO_2$ (MMNs) spectra and the MWCNT/$Fe_3O_4$@$SiO_2$ nanocomposite spectra was associated with the intensity of the bands.

The X-ray diffractogram (Figure 3B) of the $Fe_3O_4$@$SiO_2$ nanoparticles revealed well-defined peaks located around $2\theta = 30.4°$, $36.02°$; $43.6°$; $53.8°$; $57.4°$; $62.9°$, characteristics of the crystalline planes (220), (311), (101), (422), (511), and (440), respectively, of the cubic structure of magnetite [33], according to JCPDS number 36–0398 [34].

These planes are associated with a spinel-like cubic structure, which shows the existence of the crystalline phase of the $Fe_3O_4$ [32,35]. On the other hand, the MWCNT XRD peaks around the graphite structure (002) and the graphite $2\theta = 26.4°$ and $44.11°$ were attributed to the graphite structure (002) and the MWCNT planes (100), respectively [36]. The diffraction pattern for the MWCNT/$Fe_3O_4$@$SiO_2$ showed a diffraction peak around $2\theta = 26°$, corresponding to the peak of the MWCNT, and the absence of the $SiO_2$ peak in the XRD pattern in the composite was due to its amorphous structure coated on the $Fe_3O_4$ nanoparticles [37]. The diffraction peak at $2\theta = 26°$ is a typical Bragg peak of pristine MWCNT and can be associated with the reflection of graphite [38].

The Mössbauer spectrum of the magnetite (Figure 4) showed two sextets: one of them is associated with the contribution of the hyperfine magnetic interaction of the $Fe^{3+}$ ions at the A sites, and the other with the coupling $[Fe^{3+} \leftrightarrow Fe^{2+}]$ in the coordination interstices of the B sites of the inverse spinel structure ($AB_2O_4$). The correlation of hyperfine parameters indicates the presence of an oxidized magnetite phase and a small fraction of hematite. The hyperfine parameters of the isomeric displacement of $0.21 \times 10^{-3}$ and $0.63 \times 10^{-3}$ m $s^{-1}$ and of the hyperfine field of 48.25 T and 43.84 T were outside the values for pure magnetite. This indicates that the material analyzed was oxidized magnetite, that is, non-stoichiometric magnetite [35]. A non-stoichiometric substance is composed of a variable structure, but it is similar to the basic structure, in this case, the magnetite can be written as $(Fe^{3+})_A(Fe_2^{2,5+})_BO_4$, where the B site represents the ions $Fe^{3+}$ and $Fe^{2,5+}$, both in octahedral coordination. The $Fe^{2,5+}$ ion receives this terminology due to the rapid electronic recombination of the ions when they were above the Verwey transition (T = 120 K) [33,39]. According to the literature, part of the synthesized magnetite may be being oxidized to maghemite, which according to Cornel would be the intermediate phase of transformation from magnetite to hematite. The conversion of magnetite in maghemite occurs with the oxidation of $Fe^{2+}$ to $Fe^{3+}$, the latter in turn, is more easily ejected out of the structure due to its smaller ionic radius. Upon contact with oxygen in the air, $Fe^{2+}$ is reduced, giving rise to $Fe_2O_3$ [40].

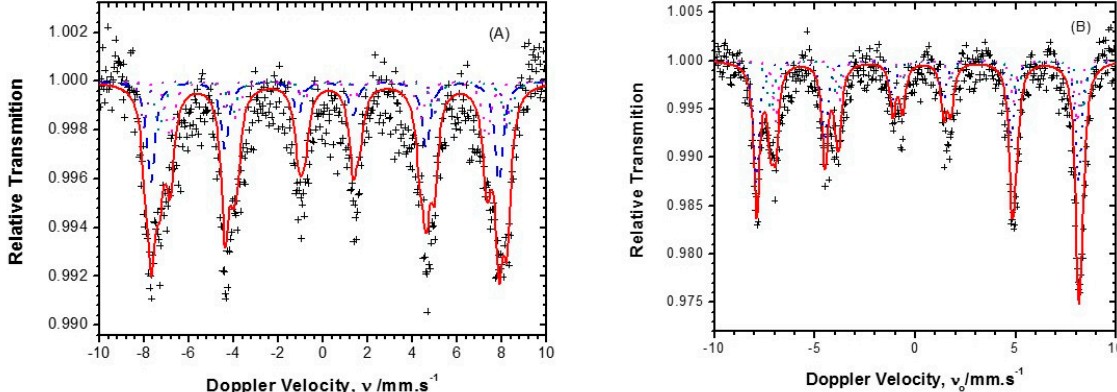

**Figure 4.** Mössbauer spectra of (**A**) $Fe_3O_4$ and (**B**) $Fe_3O_4@SiO_2$.

The magnetite sub specters were also observed in the $Fe_3O_4@SiO_2$ sample, as shown in Figure 4B. The difference from the previous sample is that the hyperfine parameters of hematite were not detected. The hyperfine parameters obtained were similar to those acquired for the sample of pure magnetite. This indicates that the functionalization process of the magnetite core with the silica coating did not cause considerable changes in the magnetite structure [35].

The point of zero charge of the $Fe_3O_4@SiO_2$ was 6.3 (Figure 5). This value is in agreement with the data found in the literature, suggesting that below this value, the surface of the MMNs acquires positive charges and above 6.3, there are predominantly negative charges [36]. As the electrode was immersed in an electrochemical solution (HCl, pH 3.0), it can be inferred that the charges present on the $GCE/MWCNT/Fe_3O_4@SiO_2$ electrode surface were mostly positive. There was an affinity between the modified electrode surface and TC.

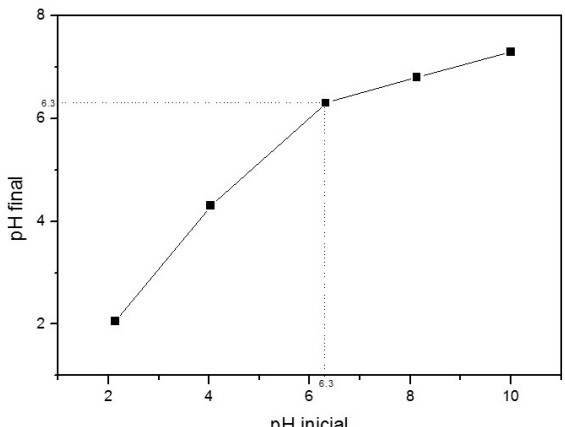

**Figure 5.** Determination of the point of zero charge of the $Fe_3O_4@SiO_2$.

### 3.2. Study of the Configuration of the Working Electrode

Figure 6 shows the study of the working electrode configuration in TC determination. The electrode modified only with $Fe_3O_4@SiO_2$ could not adhere to the electrode surface stably. The electrode modified with MWCNTs showed no TC signal (oxidation or reduction) and showed a high capacitive current. The $GCE/MWCNT/Fe_3O_4@$ showed an anodic peak current at +0.52 V and a cathodic peak current at +0.50 V. The proposed electrode showed better performance in determining TC due to the stability of the sensor, high electrical conductivity, and the increase in the surface area of the electrode [41]. The use of the Randles–Sevcik Equation (1), confirms the increase in the electroactive area of the

proposed electrode in comparison to the base electrode (GCE). The electroactive area was estimated using a solution of $1.0 \times 10^{-3}$ mol $L^{-1}$ $K_3[Fe(CN)_6]$ in 0.1 mol $L^{-1}$ KCl.

$$Ip = 2.69 \times 10^5 \, A \, D^{1/2} \, n^{3/2} \, v^{1/2} \, C \qquad (1)$$

where Ip is the peak current and D is the diffusion coefficient of $K_3[Fe(CN)_6]$ $(7.6 \times 10^{-6} cm^2 \, s^{-1})$.

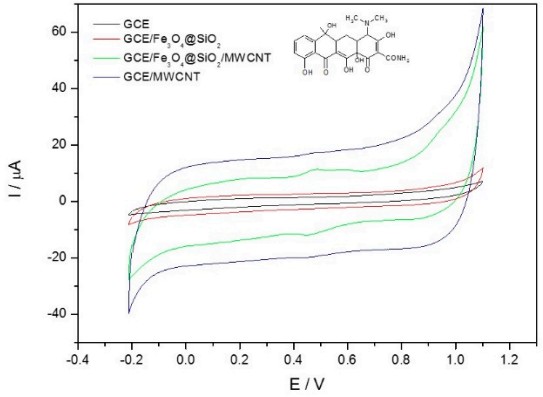

**Figure 6.** Cyclic voltammograms obtained using different electrodes: GCE; $GCE/Fe_3O_4@SiO_2$, GCE/MWCNT, and $GCE/MWCNT/Fe_3O_4@SiO_2$. Measurements performed in the presence of 22.0 $\mu$mo $L^{-1}$ of tetracycline in $1 \times 10^{-3}$ mol $L^{-1}$ HCl at pH 3.0. Scan rate: 25 mV $s^{-1}$.

The electroactive area was 0.089 $cm^2$ for GCE and 0.158 $cm^2$ for $GCE/MWCNT/Fe_3O_4@SiO_2$. The results indicated that the MWCNT not only stabilized the nanoparticles of the $Fe_3O_4@SiO_2$ on the electrode surface, but also that these materials together presented a greater surface area.

### 3.3. Electrochemical Behavior of the Sensor

The electrochemical behavior of the proposed sensor was investigated by the cyclic voltammetry technique in different scan rates. The plot of log $I_{pa}$ (anodic peak current) versus log $v$ (scan rate) (Figure 7B) yielded a straight line, with a slope of 0.94. Therefore, the process is controlled by adsorption, since for processes controlled only by diffusion, the slope must be 0.5 [42,43].

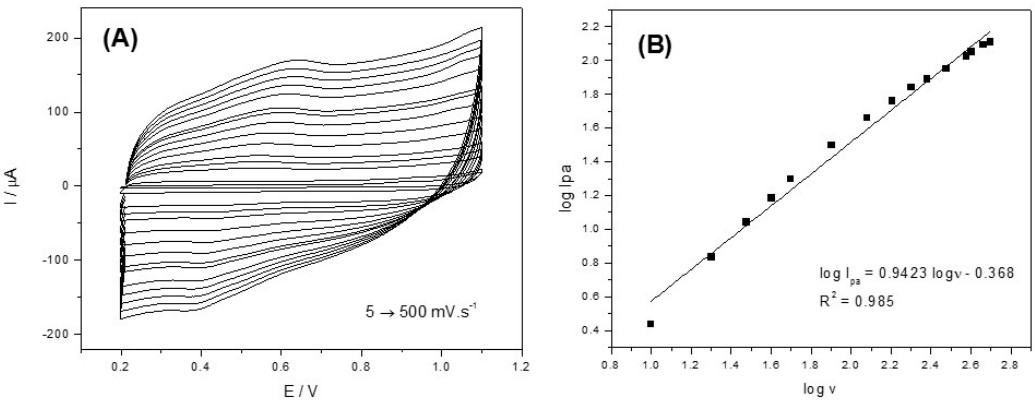

**Figure 7.** (**A**) Cyclic voltammograms of $GCE/MWCNT/Fe_3O_4@SiO_2$ at the presence of 22.0 $\mu$mo $L^{-1}$ of the TC in HCl, pH 3.0, 10.0 to 500 m V $s^{-1}$; (**B**) log $I_{pa}$ vs. log $v$.

### 3.4. Optimization of Experimental Parameters for GCE/MWCNT/Fe₃O₄@SiO₂ Sensor

Figure 8A,B shows the influence of the modifying materials in the surface of GCE. For the MWNCTs, at a concentration of 2.50 mg mL$^{-1}$, there was a significant increase in the current as smaller amounts of MWNCTs probably did not stabilize the nanocomposite of Fe₃O₄@SiO₂. For the Fe₃O₄@SiO₂, the best response was obtained using 2.5 mg mL$^{-1}$; smaller amounts resulted in low electron transfer. The behavior of the proposed sensor was evaluated at different pH values (Figure 8C). The results obtained showed that at pH 3, there was the highest anodic current intensity, thus, pH 3 was chosen as the optimal value. The protonation of dimethyl amino group present in the TC is facilitated in an acidic medium [44,45]. The specific interactions of hydrogen in the reduced quinone form could explain the difference in TC oxidation for pH values below 4.0. Strong intramolecular hydrogen interactions can take place due to the high acidity of the environment, resulting in greater sensitivity [41]. Figure 8D presents the results of the study of the support electrolyte influence on the detection of TC. Hydrochloric acid showed a significant increase in analytical signal in relation to the electrolytes evaluated. HCl did not show any detectable interaction with TC, which could lead to a decrease in analytical signal. Thus, the electrolytic medium selected was HCl.

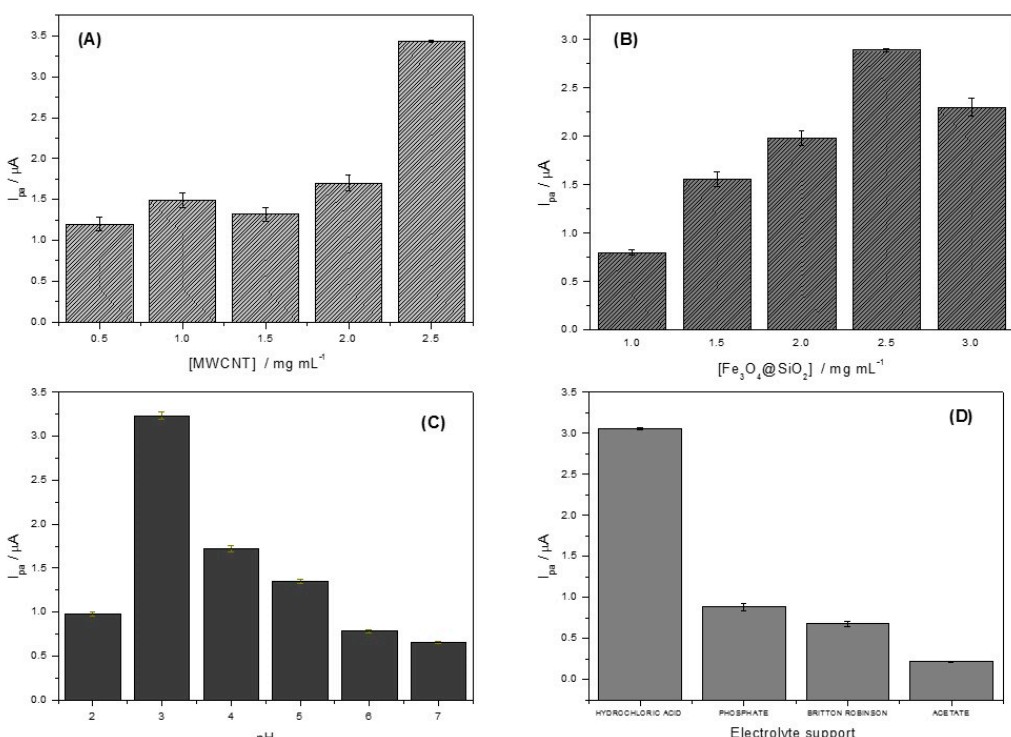

**Figure 8.** Optimization of the electrochemical parameters of the sensor. Effects of (**A**) amount of MWCNT and (**B**) amount of Fe₃O₄@SiO₂, (**C**) pH, and (**D**) composition of electrolyte support. Scan rate 25 mV s$^{-1}$ and potential range between 0 and 0.40 V.

### 3.5. Influence of Electrochemical Technical

Table 1 presents the results of the evaluation of the electrochemical techniques used. According to the results, AdSDPV showed the highest sensitivity in TC detection, and therefore was used in later studies.

**Table 1.** Influence of electrochemical techniques in the determination of TC under optimized conditions ($1 \times 10^{-3}$ mol L$^{-1}$ HCl pH 3.0).

| Technical | Sensitivity/µAmol L$^{-1}$ |
|:---:|:---:|
| AdSDPV | 0.51 |
| DPV | 0.14 |
| SWV | 0.0037 |

### 3.6. Analytical Curve

Figure 9 illustrates the voltammograms obtained with successive additions of TC in concentrations between 4.0 and 64 µmo L$^{-1}$. The voltammograms presented two TC oxidation peaks, which may be associated with the dimethyl ammonium and phenol groups present in the TC. The first peak exhibited one linear region and the second peak presented two linear regions. The second peak was chosen to obtain the analytical curve (Equations (2) and (3)).

$$I_{pa} = 41.71 + 0.55 \times [TC] \qquad r = 0.997 \qquad (2)$$

$$I_{pa} = 50.78 + 0.29 \times [TC] \qquad r = 0.994 \qquad (3)$$

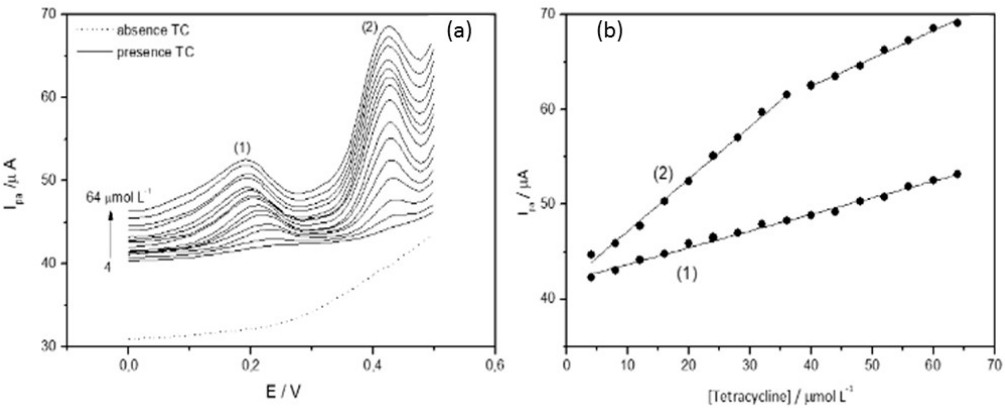

**Figure 9.** (**a**) Voltammograms for the detection of TC concentrations obtained for 4.0 to 64 µmo L$^{-1}$. (**b**) Analytical curve constructed from voltammograms (Figure 9a). Measurements obtained in $1 \times 10^{-3}$ mol L$^{-1}$ HCl at pH 3.0.

The detection limit (LOD) and the quantification limit (LOQ) were 1.67 and 4.0 µmo L$^{-1}$, respectively (obtained from Equation (2)).

Reproducibility between different electrodes (*n* = 2) was investigated in the presence of TC 31.0 µmo L$^{-1}$. The results obtained showed a low relative standard deviation (RSD), −0.71%, showing the reproducibility of the method. Table 2 shows the comparison of the method proposed in this work with sensors described in the literature for TC detection. The GCE/MWCNT/Fe$_3$O$_4$@SiO$_2$ sensor presented a low LOD when compared to other sensors, showing the advantage of being a simple and reproducible method.

**Table 2.** Comparison of the proposed sensor with other methods described in the literature.

| Electrode | Linear Range (µmol L$^{-1}$) | LOD (µmol L$^{-1}$) | Reference |
|---|---|---|---|
| GR-Pol [a] | 3.0–95.0 | 2.6 | [5] |
| CB-FB/GCE [b] | 5.0–120 | 1.15 | [46] |
| MIOPPy-AuNP/SPCE [c] | 1.0–20 | 0.65 | [45] |
| PtNPs/C/GCE [d] | 9.99–44.01 | 4.28 | [41] |
| GCE/MWCNT/Fe$_3$O$_4$@SiO$_2$ | 4.0–36 | 1.67 | This work |

[a] Polyurethane-graphite composite electrode; [b] Glassy carbon electrode modified with potato starch and black carbon; [c] Printed electrode modified with molecularly printed polypyrrole and gold nanoparticles; [d] Glassy carbon electrode modified with platinum nanoparticles.

### 3.7. Selectivity

The selectivity of the GCE/MWCNT/Fe$_3$O$_4$@SiO$_2$ sensor was evaluated in the presence of three different substances. Doxycycline (DC) has a similar structure to TC; diclofenac (DF), which consists of electroactive functional groups; and amoxicillin (AM), which is an antibiotic structurally different from TC. As illustrated in Figure 10, DC presented a greater analytical response compared to AM and DF; this behavior is due to the structural similarity with TC. Although the three interferents evaluated presented an analytical response in the same potential range when interacting with the proposed sensor, these compounds did not make it impossible to detect TC.

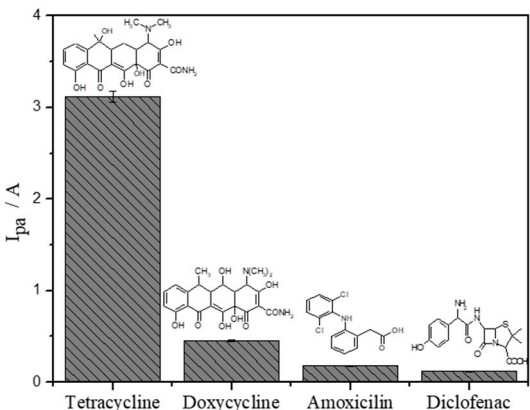

**Figure 10.** Selectivity study of the proposed sensor in the presence of 22 µmo L$^{-1}$ TC in $1 \times 10^{-3}$ mol L$^{-1}$ HCl at pH 3.0.

The modification of the electrode surface with MWCNT and Fe$_3$O$_4$@SiO$_2$ facilitated the process of electronic transfer. The modifiers provided increased conductivity and electroactive area, providing better interaction with TC. The optimization of experimental parameters such as the amount of modifiers, pH, and electrolyte support also favored the selective detection of TC. The entire optimization process was carried out to improve the oxidation conditions of the TC on the surface of the proposed electrode, thus the method resulted in greater selectivity for the TC.

### 3.8. Application of the Method

Tables 3 and 4 show the results obtained in the application of the proposed sensor in the milk and river water samples enriched with TC. The proposed method under optimized conditions showed high recovery values (between 91 and 117%) and low RSD (between 0.1 and 1.8). Additionally, the developed sensor did not present a matrix effect.

**Table 3.** Determination of TC in water samples.

| River Water Sample | Tetracycline/$\mu$mol L$^{-1}$ | | Recovery (%) |
| --- | --- | --- | --- |
| | Added | Detected * | |
| No. 1 | 5.2 | (6.1 $\pm$ 0.8) | 117.9 |
| No. 2 | 9.2 | (8.7 $\pm$ 0.4) | 94.9 |

* Average three determinations.

**Table 4.** Determination of TC in milk samples.

| Milk Sample | Tetracycline/$\mu$mol L$^{-1}$ | | Recovery (%) |
| --- | --- | --- | --- |
| | Added | Detected * | |
| No. 1 | 10 | (10 $\pm$ 0.2) | 102.0 |
| | 20 | (20.1 $\pm$ 1.2) | 100.3 |
| No. 2 | 10 | (9.1 $\pm$ 0.1) | 91.0 |
| | 20 | (21.8 $\pm$ 1.8) | 108.8 |

* Average three determinations.

## 4. Conclusions

The developed sensor, GCE/MWCNT/Fe$_3$O$_4$@SiO$_2$, showed excellent performance in TC detection due to the high electrical conductivity and surface area provided by the modifiers. The modifying materials were properly characterized. The experimental and operational parameters of the sensor were optimized, providing a simple and reproducible method with high sensitivity and selectivity. The results obtained in the application of the sensor in real samples of milk and water showed high recovery rates, which shows that the proposed method is a viable alternative in TC detection in this type of sample.

**Author Contributions:** E.F.A. developed and applied the methodology and wrote the manuscript; D.N.d.S. contributed to the writing of the manuscript; M.C.S. contributed to the methodology and characterization of the materials; A.C.P. contributed to the development of the entire analytical method. All authors have read and agreed to the published version of the manuscript.

**Funding:** This research received no external funding.

**Institutional Review Board Statement:** Not applicable.

**Informed Consent Statement:** Not applicable.

**Data Availability Statement:** Not applicable.

**Acknowledgments:** The authors are grateful to Fapemig, INCT-DATREM, CNPq, Capes for financial support and Lucas Franco Ferreira, FQMat, and Labsensor.

**Conflicts of Interest:** The authors declare no conflict of interest.

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
