# Peer review of "Development of an Electrochemical Sensor Based on Nanocomposite of Fe3O4@SiO2 and Multiwalled Carbon Nanotubes for Determination of Tetracycline in Real Samples"

_2673-3293, doi:10.3390/electrochem2020018_

Round 1

Reviewer 1 Report

This paper presented an electrochemical sensor based on MWCNT/Fe3O4@SiO2 modified glass carbon electrode for the detection of tetracycline. The properties of the prepared composite material were characterized by a series of methods including XRD, FTIR, Mössbauer Spectroscopy and SEM. The experimental designs and overall expression of the manuscript are scientifically sound. However, there is not sufficient novelty for this work. Besides, the analytical performance of the sensor towards tetracycline is not superior to other works that could achieve LOD at ng/mL level. There are some questions to be solved.

  1. In Line 230, Page 7, the value 0,158 cm2 doesn’t seem to be correct.
  2. The detection mechanism using MWCNT/Fe3O4@SiO2 towards tetracycline remains unclear. Why does the sensor have specificity for tetracycline? The authros claimed this behavior is due to the structural similarity. I suggest more convincing discussion should be provided.
  3. The authors should revise some grammatical and syntax errors carefully.

Reviewer 2 Report

Interesting paper that in my modest opinion deserves to be published, after minor revisions concerning the english language.

Reviewer 3 Report

Authors reported an electrochemical sensor (GCE/MWCNT/Fe3O4@SiO2) based on composite of multiwalled carbon nanotubes (MWCNT) and nanocomposite of Fe3O4@SiO2 (MMN) on glassy carbon electrode (GCE) which is developed for the detection of tetracycline (TC).

1. The manuscript is well-written and organized well.

2. Introduction can be improved by adding a few more Iron-based sensor materials,

https://doi.org/10.1016/j.electacta.2018.09.028

https://doi.org/10.1016/j.jelechem.2018.04.058

https://doi.org/10.1002/elan.201700750

DOI: 10.1039/C8RA08017H 

3. Impedance spectroscopy or EASA study may be added.

4.  JCPDS number must be charted near the XRD spectra.

5. Author cannot claim that Fig 2A is MWCNT. There is no evidence for it.

none of the nanotubes is found.

6. Provide the reason for the selective pH response (pH 3).

7. Also the reason for choosing HCL medium, specifically?

8. Either choose ( GCE/MWCNT/ Fe3O4@SiO2) or (Fe3O4@SiO2/MWCNT/GCE), this may confuse the readers.

Round 2

Reviewer 1 Report

The authors have replied most of the questions. The manuscript can be published in its present form.

Reviewer 3 Report

The work can be accepted in the present form.